# The Effects of Dobutamine in Septic Shock: An Updated Narrative Review of Clinical and Experimental Studies

**DOI:** 10.3390/medicina60050751

**Published:** 2024-04-30

**Authors:** Arnaldo Dubin, Matías Mugno

**Affiliations:** 1Cátedras de Terapia Intensiva y Farmacología Aplicada, Facultad de Ciencias Médicas, Universidad Nacional de La Plata, 60 y 120, La Plata B1902AGW, Argentina; 2Sanatorio Otamendi, Azcuénaga 870, Ciudad Autónoma de Buenos Aires C1115AAB, Argentina; mugnom@otamendi.com.ar

**Keywords:** dobutamine, sepsis, shock, cardiac output, blood pressure, microcirculation

## Abstract

The key objective in the hemodynamic treatment of septic shock is the optimization of tissue perfusion and oxygenation. This is usually achieved by the utilization of fluids, vasopressors, and inotropes. Dobutamine is the inotrope most commonly recommended and used for this purpose. Despite the fact that dobutamine was introduced almost half a century ago in the treatment of septic shock, and there is widespread use of the drug, several aspects of its pharmacodynamics remain poorly understood. In normal subjects, dobutamine increases contractility and lacks a direct effect on vascular tone. This results in augmented cardiac output and blood pressure, with reflex reduction in systemic vascular resistance. In septic shock, some experimental and clinical research suggest beneficial effects on systemic and regional perfusion. Nevertheless, other studies found heterogeneous and unpredictable effects with frequent side effects. In this narrative review, we discuss the pharmacodynamic characteristics of dobutamine and its physiologic actions in different settings, with special reference to septic shock. We discuss studies showing that dobutamine frequently induces tachycardia and vasodilation, without positive actions on contractility. Since untoward effects are often found and therapeutic benefits are occasional, its profile of efficacy and safety seems low. Therefore, we recommend that the use of dobutamine in septic shock should be cautious. Before a final decision about its prescription, efficacy, and tolerance should be evaluated throughout a short period with narrow monitoring of its wanted and side effects.

## 1. Introduction

The key objective in the hemodynamic treatment of septic shock is the optimization of tissue perfusion and oxygenation. This is achieved by the utilization of fluids, vasopressors, and inotropes. Dobutamine is the inotrope most commonly used for this purpose. A survey among 839 physicians from 82 countries has recently evaluated the current practice of inotropic treatment in shock states [1]. Dobutamine was chosen as the first-line inotrope in 84% of questionnaire respondents. Most of them (65%) identified persistent hypoperfusion (e.g., alterations in skin perfusion and oliguria) or persistent hyperlactatemia despite appropriate fluid and vasopressor administration as the trigger for inotrope use. A simultaneous international panel of experts also recommended inotropes for the treatment of septic and cardiogenic shock and considered dobutamine as the first-line agent [1]. Inadequate cardiac output and signs of tissue hypoperfusion were pointed out as indications and goals for the inotropic treatment. With a good degree of consensus, experts launched a strong recommendation for the use of inotropes in septic shock. Furthermore, the Surviving Sepsis Campaign suggested adding dobutamine to norepinephrine or administering epinephrine alone for adults with septic shock and myocardial dysfunction with persistent hypoperfusion, despite the correction of hypovolemia and arterial hypotension (weak recommendation and low quality of evidence) [2].

Despite these recommendations, with the exception of a network meta-analysis that found a reduced risk of 28-day mortality with a norepinephrine and dobutamine combination [3], there is limited scientific evidence that supports the beneficial effect of dobutamine on the outcome of patients with shock. On the contrary, some studies suggest that the drug exhibits a low profile of efficacy and safety in both cardiogenic and septic shock. In patients with severe heart failure and cardiogenic shock, its use has been associated with increased mortality [4,5]. In patients with septic shock, the administration of dobutamine was an independent predictor of 90-day mortality, even after adjustment with a propensity score for inotropic treatment [6]. Another retrospective study showed a similar detrimental effect along with a higher occurrence of atrial fibrillation [7]. A propensity-score-matched analysis also showed that hospital mortality was consistently higher in septic patients treated with dobutamine (60.2% vs. 49.4%) [8]. On top of this, some observational studies showed that dobutamine produces unpredictable cardiovascular actions and common untoward effects, such as tachycardia and arterial hypotension [9,10,11]. Overall, the results about the effects of dobutamine on the outcome of septic shock are inconclusive. The ongoing adjunctive dobutamine in septic cardiomyopathy with tissue hypoperfusion (ADAPT) trial will probably shed some light on this issue. The study is recruiting septic shock patients with a left ventricular ejection fraction ≤ 40% and a left ventricular outflow tract velocity time integral < 14 cm who are randomized to placebo or dobutamine [12]. The study is expected to be completed on 20 December 2024.

Despite the fact that dobutamine was introduced almost half a century ago for the treatment of septic shock [13], and there is widespread use of the drug, several aspects of its pharmacodynamics remain poorly understood. This narrative review is aimed at performing a reappraisal of its clinical pharmacology, especially in septic shock. Our aim is to show that the effects of dobutamine are commonly variable, heterogeneous, and highly dependent on the underlying illness.

## 2. Pharmacodynamics and Pharmacokinetics of Dobutamine

Dobutamine is clinically used as a racemic mixture of two enantiomers [14,15,16]. The pharmacodynamic activity of the racemate is the consequence of the interaction of the individual characteristics of stereoisomers. (−)-dobutamine behaves as a powerful adrenergic α_1_ agonist and vasoconstrictor with weak β_1_ and β_2_ activity in both vascular and cardiac receptors. (+)-dobutamine has a similar affinity for adrenergic α_1_ receptors but lacks intrinsic activity. So, it is an α_1_ antagonist. It additionally displays some β_2_ agonism in vascular receptors and is a powerful agonist of β_1_ and β_2_ adrenoreceptors in cardiac muscle. As a cardiac β_1_ adrenergic agonist, its effect on cardiac contractility is comparable to that of isoproterenol and norepinephrine. Thus, (+)-dobutamine mainly has vasodilatory and inotropic effects. Taking into account that a substantial portion of the effect on contractility depends on augmented cardiac α_1_ activity, dobutamine produces less tachycardia than other adrenergic agents. Given that (−)-dobutamine is a vasoconstrictor and (+)-dobutamine is a vasodilator, the opposite effects of each stereoisomer result in the lack of direct effect of the racemate on vascular tone. Consequently, the reduction in vascular tone and systemic peripheral resistance are reflex responses to the increase in cardiac output. Figure 1 displays the key pharmacodynamic and cardiovascular effects of dobutamine.

In severe heart failure, the half-life of dobutamine is about 2 min [18]. Thus, the steady state is reached after a few minutes of infusion start. Stable plasma concentrations are proportional to the infusion rate. This indicates the lack of saturation of the mechanism of elimination, which follows a first-order kinetics [19,20]. The half-life does not depend on cardiac output. Since the distribution volume is related to the extent of edema, similar infusion rates can result in variable plasma concentrations of the drug [18]. In addition, continuous infusion is associated with significant hemodynamic tolerance. The cardiovascular effects at 72 and 96 h were 66% and 57% of that at 2 h, respectively [21]. High variability in dobutamine clearance has also been reported [19]. This suggests that the dobutamine infusion rate should not be titrated to obtain predetermined plasma levels but to physiologic end points [22].

A major metabolite of dobutamine is 3-O-methyldobutamine, which is formed enzymatically by catechol-O-methyltransferase [23]. This metabolite has a longer half life, and its (+)-enantiomer is an α_1_ antagonist that might induce vasodilation [24].

## 3. Effects of Dobutamine in Normal Subjects

### 3.1. Cardiovascular Effects

Dobutamine is frequently characterized as an inodilator, a drug with inotropic and vasodilatory properties. In normal animals, however, it primarily behaves as a pure inotropic drug. This action results in increases in stroke volume and cardiac output. With increasing doses of dobutamine, the heart rate initially remains unchanged. Thereafter, higher doses induce tachycardia, which contributes to improved cardiac output. The increase in cardiac output produces elevations in blood pressure and a baroreceptor-mediated reduction in systemic vascular resistance [16,25]. In addition, dobutamine improves ventriculoarterial coupling (effective arterial elastance–end-systolic elastance ratio) by increasing contractility [26].

In healthy volunteers, an observational study assessed the effect of dobutamine infusion rates of 2.5, 5, and 10 μg/kg/min on the drug plasma concentration and hemodynamics [27]. Fifteen minutes after the start of each dose, rising steady-state concentrations were reached. As evidence of the short elimination half-life, the dobutamine plasma concentration was undetectable after 10 min of cessation of the infusion. Rising doses of dobutamine produced elevations in cardiac output, which were correlated to drug plasma levels. On the contrary, heart rate and stroke volume showed more complex relationships to dobutamine plasma levels. Heart rate remained unchanged with the lowest infusion rate, but thereafter, it sharply increased. Stroke volume increased significantly at dobutamine plasma concentrations produced by the lowest infusion rate but subsequently remained constant or even was reduced. Therefore, cardiac output and blood pressure were linearly related to the escalating doses of dobutamine. With an infusion rate of 2.5 μg/kg/min, the increase in both variables resulted from the improvement in stroke volume. Further increases with higher doses only depended on tachycardia. A similar biphasic response in stroke volume was found in patients without left ventricular wall motion abnormalities who underwent dobutamine stress echocardiography [28]. In this study, however, the inflection point appeared at an infusion rate of 10 μg/kg/min.

### 3.2. Metabolic Effects

Catecholamines have prominent metabolic actions that include a thermogenic effect. Incremental infusions of epinephrine produce stepwise increases in metabolic heat production [29]. Increases in metabolic rate and oxygen consumption (VO_2_) are associated with initial increases in the respiratory quotient, which has been interpreted as the consequence of reduced lipid oxidation and elevated carbohydrate oxidation [30]. Subsequently, the respiratory quotient normalizes and finally falls as a probable expression of increased lipolysis. In healthy volunteers, increasing doses of dobutamine also produce a marked calorigenic action, which might be attributed to complex α and β adrenergic metabolic effects [31,32]. An infusion of 10 μg/kg/min produces an increase in energy expenditure of 33% and a reduction in the respiratory quotient, from 0.85 to 0.80, along with increased plasmatic levels of glycerol and free fatty acids [31]. In contrast, another study found that the calorigenic effect was not associated with changes in the respiratory quotient [33].

## 4. Effects of Dobutamine in Cardiac Failure

In experimental models of cardiogenic shock, the effects of dobutamine mainly resemble those found in normal subjects. In dogs with myocardial infarction induced by ligating the left anterior descending coronary artery, dobutamine-induced dose-related increases in cardiac output and contractility restored arterial blood pressure and reduced total peripheral resistance [34]. In other experimental models of acute congestive heart failure, dobutamine also increased cardiac output and blood pressure [35,36,37,38,39].

The effects of dobutamine in patients with cardiac failure are somewhat different. Even though cardiac output is increased, blood pressure remains unchanged [40,41,42]. Since the elevation in cardiac output should increase blood pressure, these results suggest a direct vasodilatory effect, which is not found in normal individuals or in experimental cardiogenic shock.

## 5. Effects of Dobutamine in Septic Shock

### 5.1. Diagnostic Use for Assessing the Metabolic Response

In septic shock, the use of dobutamine is aimed at improving cardiac output and tissue oxygenation [1,2]. This hypothetical usefulness has been applied to diagnostic and therapeutic purposes. In many studies, dobutamine was utilized for the so-called oxygen flux test. This consists of an evaluation of VO_2_ behavior in response to increased oxygen delivery (DO_2_). The presence of the dependence of VO_2_ on DO_2_ has been attributed to the presence of masked oxygen debt and inadequate tissue oxygenation and was associated with higher mortality [43]. Nevertheless, this issue is complicated by the calorigenic effect of the drug, which can primarily increase VO_2_. Some studies found that escalating doses of dobutamine induces stepwise increases in VO_2_ and DO_2_, without changes in respiratory quotient [44,45,46]. In other observational studies, the lack of changes in VO_2_ in response to dobutamine-induced increases in DO_2_ was associated with worse outcomes compared to responders. Given the potential calorigenic effect of dobutamine, the explanation for these findings might be the cell’s inability to increase oxidative metabolism during sepsis, as an expression of a more severe underlying disease [47,48]. In other studies, however, dobutamine failure to increase VO_2_ was not related to either lactate levels or outcome [49,50]. Therefore, understanding of the effects on VO_2_ is unclear, and the test lacks usefulness.

### 5.2. Effects on Systemic Hemodynamics and Tissue Perfusion

Beyond the effects on the outcome, the hemodynamic actions of dobutamine in septic shock are also controversial. Experimental and clinical studies showed beneficial effects, such as increases in cardiac output and systemic DO_2_ [51,52,53]. Furthermore, improvements in splanchnic perfusion and tissue oxygenation have also been found in experimental and clinical settings [54,55,56,57]. In patients with septic shock, dobutamine increased DO_2_, along with an improvement in intramucosal acidosis and a reduction in lactate levels [56]. Other studies also showed beneficial effects of dobutamine on gastric mucosal perfusion in septic patients [55,56,57,58,59,60]. Thus, dobutamine might be useful for the recruitment of microcirculation. It avoids arteriolar constriction and preserves villus blood flow in endotoxemic rats [61] and improves jejunal mucosal microcirculation in swine with fecal peritonitis [62]. In experimental models, beneficial effects were also found in hepatic microcirculation [63,64]. In patients with septic shock, the administration of 5 μg/kg/min for 2 h improved sublingual microcirculation [65]. Remarkably, the microvascular effects were not related to modifications in systemic hemodynamics.

Although the aforementioned results suggest that dobutamine has an attractive physiological profile in septic shock, other data show that it induces unpredictable and heterogeneous effects. In some experimental and clinical studies, the effects on contractility are blunted in sepsis. The inotropic effect and the ability to increase blood pressure, but not the increase in heart rate, were diminished in those who were endotoxemic compared to control cats (Figure 2) [66]. In rats with cecal ligation and puncture-induced sepsis, dobutamine neither improved myocardial function and hemodynamics nor attenuated myocardial injury [67]. In a similar model, the impaired inotropic response was related to phosphodiesterase 4 upregulation, whereas myocardial surface expression of β1-adrenoceptors and α-subunits of three main G protein families was unaltered [68].

In sheep endotoxemia, it was shown that dobutamine increased cardiac output but decreased the fraction of flow directed to the superior mesenteric artery [69]. Additionally, the elevation in cardiac output fully resulted from increased heart rate, since the stroke volume not only did not increase but had a trend to decrease. In addition, blood pressure and systemic vascular resistance were reduced. Considering that cardiac output was preserved, these findings mean that dobutamine mainly behaved as a vasodilator (Figure 3). In this experimental model, dobutamine induced vasodilation and tachycardia without evidence of an inotropic effect. Furthermore, gut mucosal minus arterial PCO_2_—a surrogate for tissue perfusion—was not improved. In a partial superior mesenteric artery occlusion study, dobutamine reduced the fraction of cardiac output directed to the gut and the intramucosal pH and augmented the portal venous–arterial lactate gradient [70]. In another model of mesenteric ischemia and reperfusion, dobutamine increased cardiac output, superior mesenteric artery blood flow, and gastric and rectal microvascular blood flow. In spite of this, the microvascular blood flow of the jejunal mucosa decreased, whereas the mucosal expression of endothelin-1 and leukocytic infiltration increased [71]. In pigs with fecal peritonitis, the elevation in cardiac output did not result in improvements in superior mesenteric artery blood flow or gut mucosal perfusion [72]. Therefore, several preclinical studies failed to show beneficial effects on either systemic cardiovascular variables or tissue perfusion.

In patients with septic shock, a large body of evidence shows that the use of dobutamine is frequently associated with the occurrence of severe vasodilation. This side effect was manifest in a randomized controlled trial, in which 5–200 μg/kg/min of dobutamine were titrated to obtain supranormal values of DO_2_ and VO_2_. This treatment not only was associated with higher mortality but also with larger requirements of norepinephrine compared to the control group [73]. Consequently, the maximal doses of norepinephrine were 1.20 vs. 0.23 μg/kg/min, respectively. In three large controlled randomized trials, in which dobutamine was used as part of the early goal-directed therapy, untoward effects related to dobutamine were not directly reported [74,75,76]. This treatment, however, was associated with higher requirements of vasopressors. This finding might be explained by the presence of dobutamine-induced vasodilation.

Some clinical studies evaluated individual responses to dobutamine with a special focus on the side effects. They found heterogeneous cardiovascular responses and frequent unwanted actions. In one of them, 19 trials of escalating doses of dobutamine were performed in 12 patients [9]. In 12 cases, the infusion was stopped because hypotension or tachycardia arose. Moreover, the stroke volume remained unchanged in most of the patients. Another study also found heterogeneous hemodynamic cardiovascular effects [10]. Most of the patients developed high heart rate and arterial hypotension without evidence of increased contractility. In addition, an observational study of 23 patients with septic shock evaluated the effects of increasing doses of dobutamine up to 10 μg/kg/min [11]. The cardiovascular effects were dichotomized, taking into account changes > or < 10% from baseline to the maximal doses reached. Dobutamine could only be increased to 10 μg/kg/min in eight patients because of the development of untoward effects, mainly arterial hypotension and tachycardia. Individual responses were quite variable. Cardiac output was increased in 70% of the patients. Mean arterial blood pressure was reduced in 43% of the patients and augmented in 22% of them. Heart rate increased and systemic vascular resistance decreased in most of the patients. Despite the fall in cardiac afterload, stroke volume only improved in 52% of the patients. (Figure 4). Basal hemodynamics were similar in stroke-volume responders and nonresponders. Nevertheless, patients with increased stroke volume in response to dobutamine showed a lower left ventricle ejection fraction and more frequently exhibited systolic dysfunction and severe systolic dysfunction (left ventricle ejection fraction lower than 30%), compared to nonresponders. Consequently, the changes in stroke volume induced by dobutamine were correlated with the basal left ventricle ejection fraction. Stroke-volume responders had increases in cardiac index and a trend to increase blood pressure. In contrast, nonresponders showed unchanged cardiac and decreased blood pressure. Therefore, dobutamine behaved as an inotrope in responders and as a vasodilator without inotropic effects in nonresponders (Figure 5). Accordingly, a prospective multicenter study in patients with systolic dysfunction and fluid unresponsiveness found that dobutamine significantly increased the biventricular indices of contractility [46]. Nevertheless, these effects were linked to significant decreases in arterial pressure. Moreover, dobutamine was discontinued because of poor tolerance in 66% of the patients. Poor tolerance was more common in acidotic patients and was linked to a higher need for vasopressors and mortality. In another study, dobutamine failed to improve the end-systolic elastance during the active phase of the illness. In the survivors, however, the ability to increase contractility was recovered by days 8–10 [77]. Table 1 summarizes the main cardiovascular effects of dobutamine in normal subjects and the response frequently found in septic shock.

Concerning the effects on sublingual microcirculation, some studies failed to find the beneficial effects that were previously reported [65]. In a series of patients with septic shock, there were no significant changes in the whole cohort, yet the individual responses were quite variable [11]. The changes in perfused capillary density were not dependent on changes in systemic cardiovascular variables but on the basal state of the microcirculation. Thus, a favorable response to dobutamine was found in patients with an altered microcirculation at baseline. In a controlled crossover trial, dobutamine showed deleterious effects on muscle and hepatic perfusion, lack of improvement of peripheral perfusion, and a nonsignificant trend to increase the sublingual microcirculation [78].

The explanation for the diverse effects of dobutamine in septic shock is uncertain. An explanation may be the complex hemodynamic profile of septic shock. The cardiovascular patterns result from interactions among variable components of hypovolemia, abnormalities in vascular tone, and myocardial dysfunction. Following fluid administration, most of the patients show hypotension, tachycardia, and normal or high cardiac output. Even though an adequate systemic DO_2_ is reached in most of the cases, multiple organ failure or cardiovascular collapse are common causes of death. Most nonsurvivors exhibit a persistent hyperdynamic state with progressive and refractory vasodilation [79]. Death is typically related to the inability to regulate peripheral circulation, not to low cardiac output. In this setting, alterations in cardiac function may contribute to hemodynamic instability but only occasionally are the main physiologic disorder or the leading cause of death.

At first glance, the usefulness of dobutamine should be limited to systolic myocardial dysfunction. The spectrum of myocardial disorders in sepsis, however, is heterogeneous. Systolic dysfunction is not so frequent (30%), and its presence has no relationship to increased mortality [80]. On the contrary, diastolic dysfunction is more common (48%) and is associated with outcome. Dobutamine, as with any inotrope, could worsen diastolic dysfunction. Dynamic left intraventricular obstruction, which is present in 22% of patients with septic shock and is related to higher mortality [81], could also be impaired by the use of dobutamine [82]. Acute stress cardiomyopathy is another condition that could be aggravated or triggered by inotropic drugs [83,84]. Thus, the wide range of cardiac alterations of septic shock might constitute a reason for the unpredictable effects of dobutamine.

Many other mechanisms might also play a role in explaining the variability of the cardiovascular responses to dobutamine. Sepsis is characterized by a wide hyporesponsiveness to endogenous mediators and exogenous drugs. This disorder depends on some factors, such as membrane hyperpolarization, decreased sensitivity to calcium, and alterations in receptors [85]. Adrenergic receptors are altered at different levels: changes in affinity, decreased number by internalization and down-regulation, and uncoupling of G-proteins. These abnormalities could modify not only the vasopressor but also the inotropic response to catecholamines [86,87,88,89].

Pharmacogenetics might be another factor involved in the interindividual variability of dobutamine pharmacodynamics. Genetic differences in the human cardiac β-adrenoceptor pathway could produce different responses to the treatment. Single-nucleotide polymorphisms in receptor-encoding genes could alter function, substrate binding affinity, expression, and both the up- and downregulation of receptors [90]. Cardiac β-adrenoceptor and Gs protein α-subunit have some genetic variation that might result in dissimilar responses to inotropes [91]. The β1-adrenoreceptor gene has two commonly variable sites, at amino acid positions 49 and 389. Arg389Gly human β1-adrenoreceptor gene polymorphism is the most studied. It is associated with increases in the activity of adenylate cyclase/protein kinase A and a higher effect on contractility compared with Gly389 carriers [92]. European people are more commonly Arg389 carriers. Subjects homozygous for the Arg389 β1-adrenoreceptor had a higher inotropic response to dobutamine than subjects carrying one or two copies of the Gly389 allele [93]. In homozygous Arg389, dobutamine increased plasma-renin activity, heart rate and contractility, and decreased diastolic blood pressure more potently than in homozygous Gly389 [94,95]. On the other hand, critically ill Arg389 homozygous neonates have lower heart rates in response to dobutamine than Gly389 homozygotes, without action on cardiac output [96]. Also, polymorphism in the human gene encoding the α-subunit of Gs protein could explain different effects on cardiac output, but the data are controversial [96,97]. One, a large clinical study showed that polymorphisms do not alter the effect of dobutamine on heart rate and blood pressure during stress echocardiography [98].

Although the chronotropic response to dobutamine is preserved or even increased during the aging process, the effect on contractility may be dampened [99,100,101,102]. The failure to increase myocardial glucose utilization could explain the reduction in inotropic response in older subjects [101]. Cardiac magnetic resonance imaging showed that dobutamine markedly decreased the left atrial passive emptying function and correspondingly increased the active emptying function in the elderly [103,104]. This could contribute to the lower effect on cardiac output. A recent study assessed young (26  ±  4 year) and older (68  ±  5 year) participants who received dobutamine in normothermia and hyperthermia (increase in core temperature of 1.2 °C) [105]. Interestingly, the inotropic response was impaired in hyperthermic but not in normothermic older subjects.

In terms of sex differences, experimental data showed that female mice have a lower increase in stroke volume but a higher chronotropic response than males [106]. Clinical studies also showed increased chronotropic response in women [94,107]. Nevertheless, during a stress echocardiography study, there were no gender differences in the chronotropic response to dobutamine, but women displayed higher increases in blood pressure than men [108]. A systematic review of the literature showed that dobutamine-induced takotsubo cardiomyopathy has a strong female predominance (86%), especially in the postmenopausal period [109]. According to experimental data, delayed recovery of left ventricular diastolic function after prolonged stress secondary to dobutamine administration may be a contributing factor [110].

## 6. Conclusions

Patients with septic shock commonly have variable and unpredictable responses to dobutamine. In many patients, arterial hypotension and tachycardia are the most noticeable effects, along with an absence of inotropic actions. The current body of evidence suggests a low efficacy and safety profile. In addition, there are contradictory reports about its effects on tissue perfusion. The diagnosis of myocardial systolic dysfunction might help to select patients, in whom dobutamine can produce beneficial effects. Since untoward effects are very common and therapeutic benefits are occasional, before a definitive therapeutic decision, efficacy and tolerance should be assessed during a short period, with narrow monitoring of its wanted and side effects (Figure 6).

## Figures and Tables

**Figure 1 medicina-60-00751-f001:**
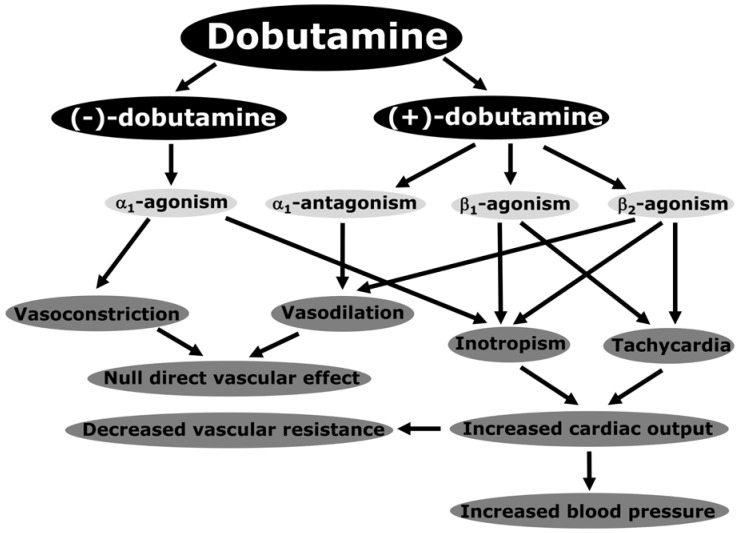
Schematic view of the main pharmacodynamic and cardiovascular effects of dobutamine. Reproduced with permission from Dubin A, Lattanzio B, and Gatti L. The spectrum of cardiovascular effects of dobutamine from healthy subjects to septic shock patients. Rev. Bras. Ter. Intensiv. 2017, 29(4), 490–498 [17].

**Figure 2 medicina-60-00751-f002:**
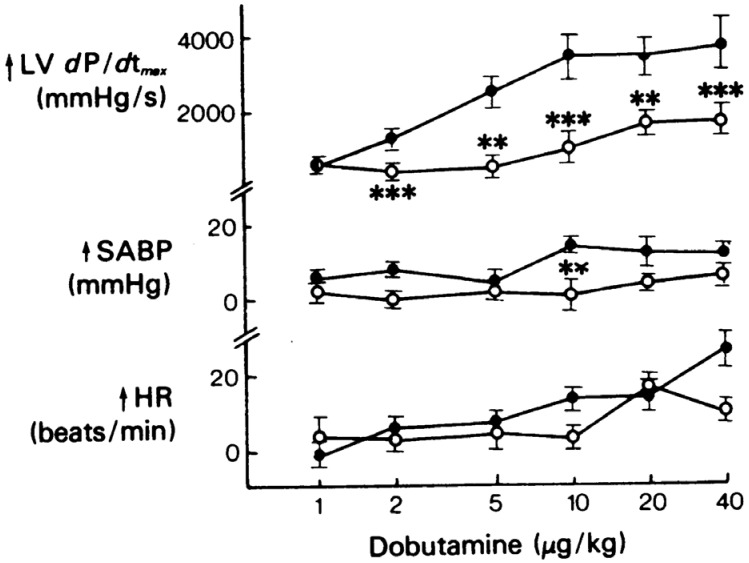
Changes in left ventricle *d*P/*d*t*_max_*, arterial blood pressure, and heart rate in response to intravenous administration of dobutamine in anesthetized cats before (•) and after (**o**) *E. coli* endotoxin. ** *p* < 0.02; *** *p* < 0.01. The effects on inotropism and blood pressure, but not on chronotropism, were diminished by endotoxin. Reproduced with permission from reference [66]. Meaning of symbols: ↑, increased.

**Figure 3 medicina-60-00751-f003:**
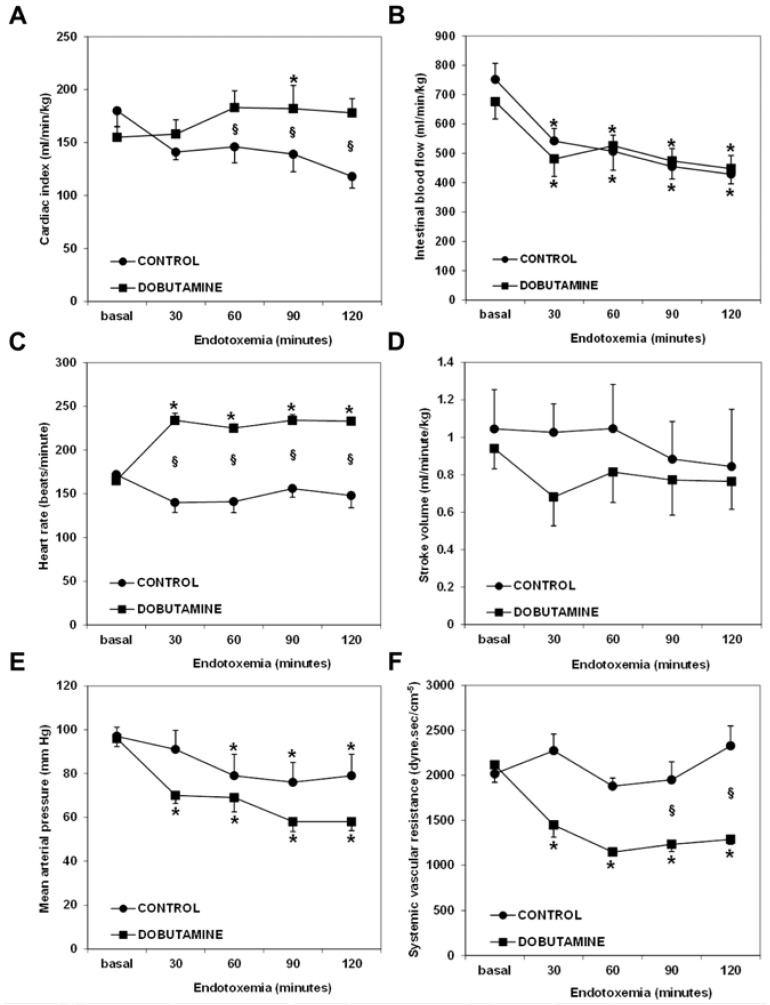
Behavior of cardiovascular variables in control and dobutamine-treated endotoxemic sheep. Panel (**A**), cardiac index. Panel (**B**), superior mesenteric artery blood flow. Panel (**C**), heart rate. Panel (**D**), stroke volume. Panel (**E**), mean arterial pressure. Panel (**F**), systemic vascular resistance. * *p* < 0.05 vs. basal; § *p* < 0.05 vs. control. The increase in cardiac output induced by dobutamine was only related to tachycardia since stroke volume had a trend to be reduced. Mean arterial blood pressure and systemic vascular resistance were reduced by dobutamine. Reproduced with permission from reference [69].

**Figure 4 medicina-60-00751-f004:**
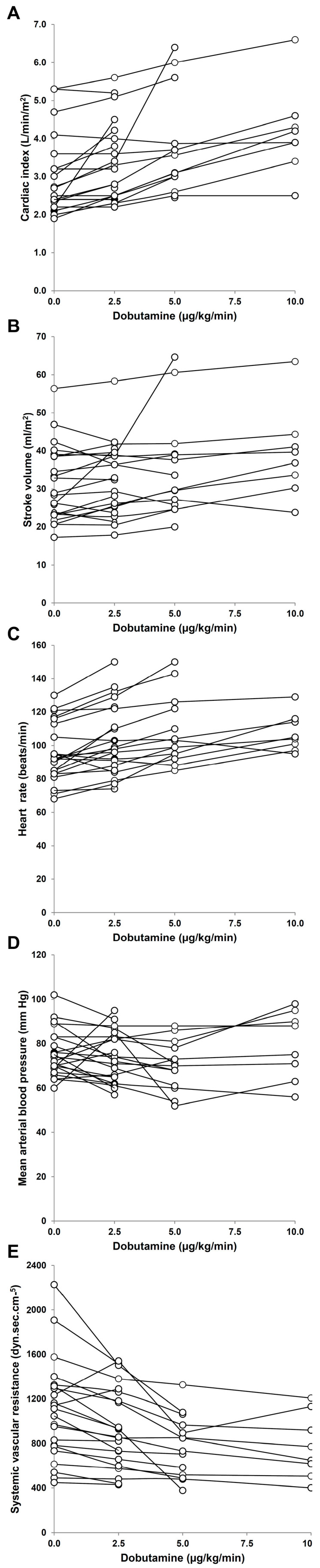
Individual responses of cardiovascular variables to increasing doses of dobutamine in patients with septic shock. Panel (**A**), cardiac index. Panel (**B**), stroke volume. Panel (**C**), heart rate. Panel (**D**), mean arterial pressure. Panel (**E**), systemic vascular resistance. Reproduced with permission from reference [11].

**Figure 5 medicina-60-00751-f005:**
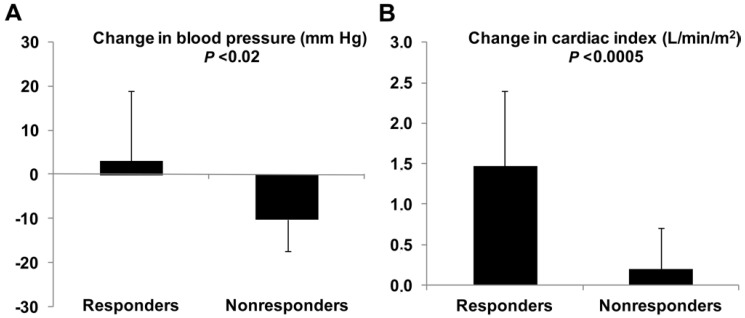
Cardiovascular behavior of stroke-volume responders and nonresponder patients with septic shock at the maximal dosage of dobutamine. In responders, dobutamine behaved as an inotrope, increasing blood pressure and cardiac index. In nonresponders, dobutamine mainly acted as a vasodilator, since blood pressure decreased and cardiac index marginally increased. Panel (**A**), change in mean arterial pressure. Panel (**B**), change in cardiac index. Built considering data from reference [11].

**Figure 6 medicina-60-00751-f006:**
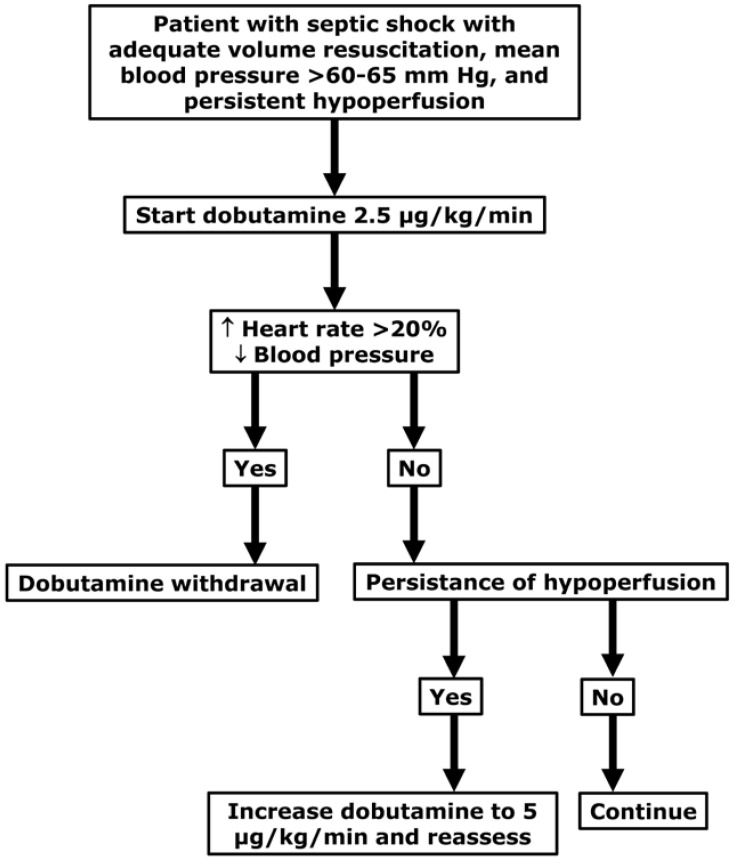
Simple algorithm for the indication and monitoring of dobutamine treatment. Meaning of symbols: ↑, increased; ↓, decreased.

**Table 1 medicina-60-00751-t001:** Summary of the cardiovascular effects of dobutamine.

	Normal Subjects and Desired Response in Septic Shock	Response Frequently Found in Septic Shock
Stroke volume	↑↑↑	↓↔
Heart rate	↑	↑↑↑
Cardiac output	↑↑↑	↑
Blood pressure	↑↑	↓
Systemic vascular resistance	↓	↓↓↓

Meaning of symbols: ↑, increased; ↓, decreased; ↔, unchanged.

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
