# Peer review of "The Effects of Dobutamine in Septic Shock: An Updated Narrative Review of Clinical and Experimental Studies"

_medicina, 2024, doi:10.3390/medicina60050751_

Round 1
Reviewer 1 Report
Comments and Suggestions for Authors
First of all, thanks for inviting me to review this manuscript called: "The Effects of Dobutamine in Patients with Septic Shock: An Updated Narrative Review".
This review is very complete from a pharmacological and physiological point of view. I have no real comments. However, maybe, some figures to explain how the monitorage of the response of dobutamine could be done, and a table summarizing the paragraph on the effect of dobutamine in septic patients could improve the global understanding of the manuscript.
Comments on the Quality of English LanguageL32 : inotroe à inotrope
Author Response
of view. I have no real comments. However, maybe, some figures to explain how the monitorage of the response of dobutamine could be done, and a table summarizing the paragraph on the effect of dobutamine in septic patients could improve the global understanding of the manuscript.
Authors’ response: To address this request, Figure 6 and Table 1 were added to the revised manuscript.
L32: inotroe.
Authors’ response: It was corrected.

Reviewer 2 Report
Comments and Suggestions for Authors
The authors summarized the effects of dobutamine. They reviewed dobutamine effects on different pathological conditions with both clinical and experimental studies. It is a valuable review since dobutamine is one of most commonly used inotropic agents. I think the title should be changed because title reflects that the review only discusses the effects of dobutamine on septic shock patients but the review includes both experimental and clinical studies.
Comments on the Quality of English LanguageThere are misspelled words in the manuscript (like inotroe in 32. line) they should be corrected.
Author Response
I think the title should be changed because title reflects that the review only discusses the effects of dobutamine on septic shock patients but the review includes both experimental and clinical studies.
Authors’ response: As suggested, the title was changed to “The Effects of Dobutamine in Patients with Septic Shock: An Updated Narrative Review of Clinical and Experimental Studies”.
There are misspelled words in the manuscript (like inotroe in 32. line) they should be corrected.
Authors’ response: The misspelling was corrected.
